# Serum Fibroblast Growth Factor 21 Levels in Children and Adolescents with Hashimoto’s Thyroiditis before and after l-Thyroxin Medication: A Prospective Study

**DOI:** 10.3390/medicina57121374

**Published:** 2021-12-17

**Authors:** Pavlos Drongitis, Eleni P Kotanidou, Anastasios Serbis, Vasiliki Rengina Tsinopoulou, Spyridon Gerou, Assimina Galli-Tsinopoulou

**Affiliations:** 12nd Department of Paediatrics, School of Medicine, Faculty of Health Sciences, Aristotle University of Thessaloniki, AHEPA University Hospital, 54636 Thessaloniki, Greece; droggpaul@yahoo.com (P.D.); epkotanidou@gmail.com (E.P.K.); tasos_serbis@yahoo.com (A.S.); vasotsino@gmail.com (V.R.T.); 2Analysi Iatriki S.A., Biopathological Diagnostic Research Laboratories, 54623 Thessaloniki, Greece; spiros.gerou@gmail.com

**Keywords:** Hashimoto’s thyroiditis, fibroblast growth factor 21, resting metabolic rate, l-thyroxin, children, adolescents

## Abstract

*Backgrounds and Objectives*: Fibroblast growth factor 21 (FGF-21) is a complex hormone, sharing common sites of action with thyroid hormones. We investigated the association among FGF-21 levels, resting metabolic rate (RMR), and l-thyroxin (LT4) treatment in children and adolescents with Hashimoto’s thyroiditis. *Materials and Methods*: A total of 60 youngsters with chronic autoimmune thyroiditis (AIT) (30 with subclinical hypothyroidism, 30 with euthyroidism) and 30 age and sex-matched healthy participants (5–18 years old) were enrolled in the study. Anthropometric, biochemical parameters, and RMR levels were assessed in all participants; serum FGF-21 levels were measured in the control group and the group with subclinical hypothyroidism before and six months after medication with LT4. *Results*: FGF-21 levels were lower in the treatment group compared with the healthy ones, but this difference was not statistically significant (*p* > 0.05); despite the increase in FGF-21 levels after six months of LT4 treatment, this difference was not statistically significant (*p* > 0.05). Free thyroxin (FT4) levels correlated well with FGF-21 levels (r = 0.399, *p* < 0.01), but further analysis revealed no interaction between these two variables. Both patient groups presented elevated triglyceride (TG) levels compared to controls (*p* < 0.05). LT4 treatment had no impact on RMR and lipid or liver or glycaemic parameters. An increase in fat mass and fat-free mass were reported, independently of FGF-21 levels. *Conclusions*: In youngsters with subclinical hypothyroidism due to Hashimoto’s thyroiditis, the serum FGF-21 levels are not significantly lower than in healthy individuals and increase after treatment with LT4 without a statistical significance. Further studies with a large number of young patients and severe hypothyroidism are recommended to confirm our results.

## 1. Introduction

Hashimoto’s thyroiditis, also referred to as chronic lymphocytic thyroiditis or chronic autoimmune thyroiditis (AIT), is a chronic autoimmune thyroid disorder characterized by increased titers of thyroid autoantibodies against thyroid peroxidase (Anti-TPOAb) and/or thyroglobulin (Anti-TgAb), lymphocytes infiltration, and thyroid tissue destruction [1,2]. AIT is a common form of thyroiditis in children and adolescents, reaching its incidence peak in early to mid-puberty and being 3–4 times more prevalent in females than males [1,3,4,5]. Hashimoto’s thyroiditis in children and adolescents is often presented at the time of diagnosis in the form of normal/subclinical hypothyroidism [6]. Treating thyroid function disorders on the grounds of AIT in pediatric individuals is essential in order to secure normal neurocognitive and somatic growth [7].

It is well documented that thyroid hormones (THs) are involved in energy and lipid metabolism, thermogenesis, and body weight control, acting on several tissues. Thus, any change in thyroid status may affect body weight and metabolic rate [8,9]. On the other hand, fibroblast growth factor 21 (FGF-21) is a complex hormone involved in energy, lipid, and glucose metabolism, sharing common biochemical pathways and sites of action with THs. FGF-21 is synthesized and acts primarily on the liver, but weaker expression has also been described in muscle, pancreas, and adipose tissue. In addition, FGF-21 acts through endocrine and paracrine mechanisms, regulating metabolic pathways such as fatty acid oxidation, glucose uptake, and thermogenesis [9,10,11,12].

Recent animal and human studies have highlighted a close bidirectional relationship between FGF-21 and THs, partially elucidated [9,13,14,15,16]. Thyroid hormones regulate the expression of the FGF-21 gene in the liver and can also increase FGF-21 levels in vivo. However, it has also been suggested that some of their key actions are largely independent [17,18,19]. Data on FGF-21 levels and their metabolic role in pediatric patients with AIT are scarce. This study aimed to measure FGF-21 serum levels in children and adolescents with Hashimoto’s thyroiditis and investigate any possible associations between FGF-21 serum levels and resting metabolic rate (RMR) and levothyroxine (LT4) treatment, or other clinical and biochemical parameters.

## 2. Materials and Methods

### 2.1. Participants

Between October 2015 and March 2020, a total of 172 children and adolescents, aged 5–18 years, were screened for AIT at the Pediatric Endocrinology Outpatient Clinic of Papageorgiou General Hospital and AHEPA University Hospital of Thessaloniki, Greece. Diagnosis of AIT was based on the presence of anti-thyroid autoantibodies (Anti-TPOAb and/or Anti-TgAb) and one or more of the following: clinical symptoms of thyroid dysfunction, goiter, or diffuse/irregular hypoechogenicity of the thyroid gland during ultrasound examination [6]. Among the 172 screened subjects, 36 with AIT (subclinical hypothyroidism) received levothyroxine (LT4) treatment [20]. A total of 6 of the 36 subjects were subsequently excluded from the study because they received medication for acute illness during their follow-up. The remaining 30 young patients comprised the “AIT treatment group”. Among them, there were 23/30 youngsters (9 boys/14 girls) with a TSH < 10 µIU/L, and 7/30 subjects (3 boys/4 girls) with TSH levels > 10 µIU/L, all presented with FT4 values within a normal range and clinical features of hypothyroidism.

From the 172 initially screened subjects, 30/172 participants (12 boys/18 girls) with AIT and euthyroidism at the time of enrolment comprised the “AIT euthyroid group”, whereas 30/172 age- and sex-matched healthy subjects (12 boys/18 girls) were also enrolled as “Control group” in the study.

All participants presented normal body mass index (BMI) for their age and sex, were drug-naive for at least 3 months, followed no special diet, and did not present any chronic and/or acute disease or menstrual disorder. The AIT treatment group was followed for six months after starting LT4 treatment.

### 2.2. Clinical and Biochemical Data

Height was measured to the nearest millimeter with a wall-mounted stadiometer (Harpenden Stadiometer, Holtain Limited, Crosswell Wales, UK). Waist, hip, and mid-upper arm circumference (MUAC) were measured with a Seca 201 measuring tape (Hamburg, Germany), and body weight was assessed with a Seca 711 scale (Hamburg, Germany). Body fat (BF) was assessed by the same experienced investigator using a skinfold caliper (Harpenden Skinfold Caliper, Baty International, West Sussex, UK) and the equations proposed by Slaughter et al. [21]. Fat mass (FM), fat-free mass (FFM), FM index (FMI), and FFM index (FFMI) were calculated [22,23]. Body mass index (BMI) was calculated, and standard deviation scores (SDS) for BMI, height, and skinfolds were determined from the WHO growth charts using the LMS growth software. All subjects underwent a complete physical examination, including posterior palpation of the thyroid gland, and were classified according to their puberty, applying Marshall and Tanner criteria [24,25].

Resting Metabolic Rate (RMR) was measured after a 12 h fast with a portable indirect calorimeter (FitMateTM, Cosmed, Rome, Italy) [26], using a pediatric face mask following the protocols proposed by the study conducted by Fullmer et al. [27].

Blood samples were collected after overnight fasting, and serum levels of biochemical parameters were measured using standard methods and an ARCHITECTc 16000 clinical chemistry system (Abbott, Abbott Park, IL, USA). Concentrations of insulin, thyroid-stimulating hormone (TSH), free triiodothyronine (FT3), free thyroxin (FT4), as well as anti-thyroid peroxidase antibody (Anti-TPOAb), and thyroglobulin antibody (Anti-TgAb) titers were measured with an ADVIA Centaur XPT Immunoassay System (Siemens Healthcare GmbH, Erlangen, Germany). Laboratory’s reference range for TSH, FT4, and FT3 levels was 0.80–3.99 μIU/L, 10.55–20.72 pmol/L, and 4.21–7.57 pmol/L, respectively. The positive cut-off value of Anti-TPOAb and Anti-TgAb titers was >60 IU/mL. A thyroid gland ultrasound was performed by the same radiologist at the beginning of the study.

Serum FGF-21 levels were measured in patients with subclinical hypothyroidism and the control group. FGF-21 levels were determined in pg/mL using the Solid Phase Sandwich ELISA method according to the manufacturer’s protocol (Quantikine^®^ Elisa, Human FGF-21 immunoassay DF 2100, R&D Systems Europe Ltd., Abingdon Science Park, Abingdon, UK) with a sensitivity of 8.69 pg/mL, intra-assay CV < 4%, inter-assay CV < 5% and an assay range of 31.3–2000 pg/mL.

In order we have indirect information regarding the dietary state in participants, all participants, with the help of their parents and/or caregivers, completed the KIDMED questionnaire at their first visit. KIDMED questionnaire consists of 16 diet-related questions. A total score of 0–3 reflects a poor adherence to the Mediterranean diet, 4–7 an average compliance, and a score of 8–12 a suitable adherence [28,29].

### 2.3. Statistical Analysis

Statistical analysis was performed using IBM SPSS Statistics version 23.0 (SPSS Inc., Chicago, IL, USA). Continuous variables were tested for normal distribution by the Kolmogorov–Smirnov or Shapiro–Wilk test. Data are presented as mean ± standard deviation (SD) or medians with lower or upper quartiles. The differences between the examined groups of individuals were investigated using the Kruskal–Wallis test and ANOVA within the GLM function with Box-Cox transformation of the response variable to improve normality. The differences between the two groups of individuals were assessed using the Tukey post-hoc and Fisher least significant difference tests. When data were paired (e.g., before vs. after), we accounted for subject ID in our model. The correlations between two continuous variables were investigated using the Spearman rank correlation test. The level of statistical significance was set at *p* < 0.05.

## 3. Results

### 3.1. Baseline Characteristics of All Studied Groups

Baseline characteristics of all studied individuals are presented in detail in Table 1.

The total study population was grouped into three groups: the patient group with subclinical hypothyroidism (AIT treatment group, *n* = 30), the patient group with AIT and euthyroidism (AIT euthyroid group, *n* = 30), and the healthy group with no AIT and normal thyroid function (control group, *n* = 30). Among all three groups, no significant differences in age, sex, Tanner stage, weight, height, standard deviation score for height (SDS Ht), BMI, %BF, FM, FFM, FMI, FFMI, waist circumference, hip circumference, and MUAC were identified (*p* > 0.05). Patients with euthyroidism presented with a lower SDS BMI, but in the post-hoc analysis, that difference remained significant only between the patients with euthyroidism and subclinical hypothyroidism (*p* < 0.05).

Serum FGF-21 levels were found not significant lower in the AIT treatment group compared to the control group (182.71 pg/mL (169.32–234.55) vs. 217.36 (193.60–235.21) pg/mL), (*p* = 0.717). FGF-21 levels presented no difference between boys and girls (*p* > 0.05) in the total study population. After adjusting for sex, Tanner stage, TSH, FT3, FT4, triglycerides (TG), total cholesterol (TC), high-density lipoprotein (HDL), low-density lipoprotein (LDL), glucose, insulin, FM or FFM, the difference of FGF-21 serum levels between the AIT treatment and the control groups did not reach statistical significance (all *p* > 0.05).

A ROC curve analysis for FGF-21, TG, and TSH in the AIT treatment and control group was also performed in order to explore further if only FGF-21 and TG compared to TSH could serve as relative sensitive markers of peripheral hypothyroidism (Figure 1). The FGF-21 and TG ROC curves were far from the diagonal. The area under the curve (AUC) of FGF-21, AUC of TG, and AUC of TSH were different. Overall, the AUC of TSH was higher than the curves of TG or FGF-21 (0.898 vs. 0.661 vs. 0.602, respectively). The optimal cut-off point was 183.07 pg/mL for FGF-21 and 0.62 mmol/L for TG. The AUC of FGF-21 did not reach significance (*p* = 0.1853). Thus, TG and FGF-21 are found as much less sensitive markers than TSH (56.67 vs. 53.33 vs. 93.33, respectively) in the context of hypothyroidism.

**Figure 1 medicina-57-01374-f001:**
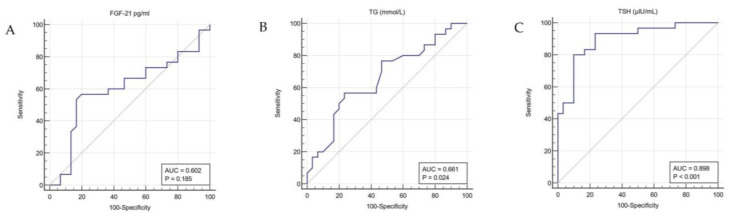
ROC curves for FGF-21 (**A**), TG (**B**), and TSH (**C**) in the AIT treatment and control groups. FGF-21: fibroblast growth factor 21; TG: triglyceride; TSH: thyroid-stimulating hormone; AIT: chronic autoimmune thyroiditis.

Serum FGF-21 levels were significantly and positively correlated with FT4 levels in the total study population (r = 0.399, *p* < 0.01) (Figure 2); this positive association between the serum FGF-21 and FT4 levels was also strongly observed in the AIT treatment group (r = 0.385, *p* < 0.05).

**Figure 2 medicina-57-01374-f002:**
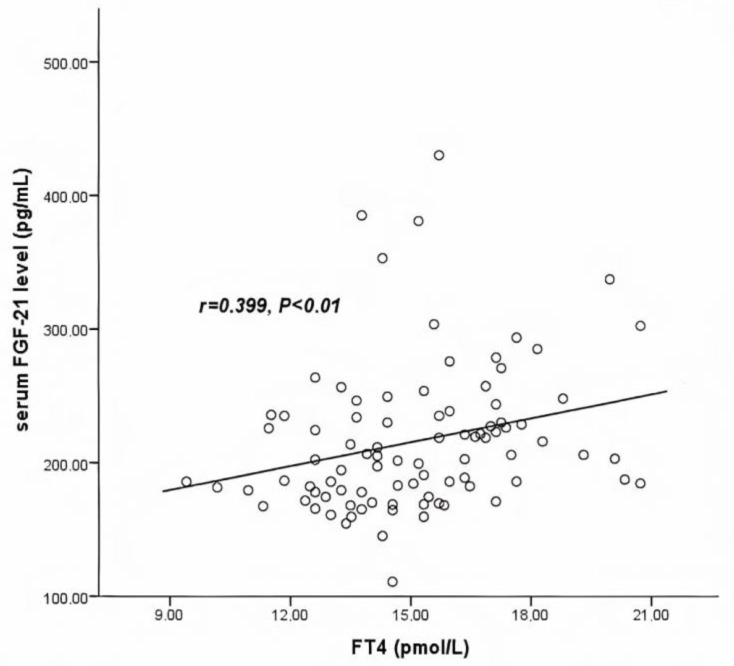
Relationships between serum FGF-21 and FT4 levels in control and AIT treatment groups before and after treatment (*n* = 90). Spearman correlation coefficients were as follows: r = 0.399, *p* < 0.01. FGF-21: fibroblast growth factor 21; FT4: free thyroxine.

However, further analysis of variance showed no actual interaction between these two variables (*p* = 0.301). In addition, serum FGF-21 levels did not correlate with any other anthropometric parameters or laboratory findings (lipidemic profile, glycemia, or liver function parameters) (*p* > 0.05) (Table 2).

Triglyceride (TG) levels were significantly higher in patients with subclinical hypothyroidism and euthyroidism compared to healthy participants (*p* < 0.05) in protocol baseline (Figure 3). However, among the two patient groups, the further pairwise comparison revealed that TG levels did not differ between the subclinical hypothyroid and the euthyroid individuals (*p* > 0.05).

Youngsters with subclinical hypothyroidism had lower RMR levels than those of the control and AIT euthyroid groups. Still, that difference was not significant after adjusting for FFM or body weight (all *p* > 0.05).

Although there were no differences between groups in the overall KIDMED score, more healthy subjects scored a suitable adherence to the Mediterranean diet than the patients (control group: 29.7% vs. AIT euthyroid group: 8.1% and AIT treatment group: 16.7%). In addition, healthy individuals consumed healthier eating compared to the patients. Indeed, control subjects reported rarer consumption of fast foods (16.7% vs. 35%), commercially baked foods (43.3% vs. 56.6%), sweet candies (56.7% vs. 75%), and on the other hand, more frequent consumption of nuts (46.7% vs. 28.3%), and a second daily serving of vegetables (33.3% vs. 18.3%) compared to patients.

### 3.2. Characteristics after LT4 Medication

LT4 therapy was initiated after baseline evaluation in participants with AIT and subclinical hypothyroidism. After 6 months of LT4 therapy, mean FGF-21 levels were not significantly increased from 182.71 pg/mL (169.32–234.55) to 198.43 pg/mL (183.86–248.42) (Table 3). Serum FGF-21 levels were not correlated with the FT4 levels after 6 months of therapy (r = 0.246, *p* > 0.05). After LT4 treatment, RMR levels did not increase significantly after adjusting for FFM or body weight (*p* > 0.05).

Both FM and FFM were increased after LT4 treatment in patients with subclinical hypothyroidism (FM after treatment: 18.62 kg (10.83–29.01) vs. FM before treatment: 8.86 kg (7.35–16.69), FFM after treatment: 57.81 kg (43.19–64.22) vs. FFM before treatment: 30.57 kg (25.64–35.30), p = 0.001 and p = 0.000, respectively). The increase in FM and FFM after LT4 therapy in the AIT treatment group was independent of the variation in serum FGF-21, BF, and BMI levels and remained significant after adjusting for age (FMI and FFMI *p* = 0.001 and *p* = 0.000, respectively).

## 4. Discussion

In this prospective study, serum FGF-21 levels were not significantly lower at baseline in children and adolescents with AIT and subclinical hypothyroidism, and they were not significantly increased after 6 months of LT4 therapy. FGF-21 showed low sensitivity as a possible marker of peripheral thyroid function and can not be used as such. FT4 concentrations correlated well with FGF-21 levels, but there were no actual interactions between these two variables. An increase in FM and FFM was reported after LT4 treatment in patients with subclinical hypothyroidism. Both AIT treatment and AIT euthyroid groups initially presented elevated TG levels compared to the control group.

Serum FGF21 levels variations are found in accordance with the previously reported levels in the current literature [12,30,31,32]. A large inter-individual variation in FGF-21 levels has been described [12]. Baseline serum FGF-21 levels were found not significantly lower in subjects with subclinical hypothyroidism than healthy ones, whereas 6 months of LT4 therapy did not significantly increase FGF-21 levels. The fact that this increment failed to reach statistical significance (*p* = 0.734) could be attributed to the subclinical hypothyroidism that the study population exhibited and the prompt initiation of LT4 replacement therapy. A more clear trend was observed in the study of Wang et al. [15], where adults with overt hypothyroidism presented decreased plasma FGF-21 levels compared to controls and subjects with subclinical hypothyroidism. In this study, plasma FGF-21 levels of the group with hypothyroidism were measured significantly higher after LT4 treatment compared to baseline. However, in another study focused on adults, mean plasma FGF-21 concentrations were significantly higher in subjects with overt hypothyroidism than in subjects with either euthyroidism or subclinical hypothyroidism [14]. Discrepancies among different reports can be attributed to differences in study design, population age ranges, the severity of hypothyroidism, and technical aspects of different FGF-21 Elisa kits used for plasma or serum FGF-21 measurements.

Dyslipidemia in the context of hypothyroidism constitutes another source of FGF-21 levels discrepancy among different studies. It is well established that THs are involved in lipid metabolism, and thus, any thyroid function abnormality impairs that balance [1]. Overt hypothyroidism is often accompanied by increased TC, LDL, apolipoprotein B, lipoprotein A, and TG levels, but such alterations on lipidemic profile are usually not apparent in subclinical hypothyroidism [14]. Recent evidence shows that patients with hypothyroidism present lower TG and LDL levels than controls [33]. Furthermore, a study of 179 children and adolescents found a positive correlation between FGF-21 and TG levels in girls, concluding that elevated levels of FGF-21 and TG in girls compared with boys may be closely related [30]. Lastly, Catli et al. [34], studying healthy children and children with subclinical hypothyroidism, observed no significant differences in lipid parameters. In our study, higher serum TG levels were detected in the AIT patients (both with subclinical hypothyroidism and euthyroidism) compared with the healthy individuals (*p* < 0.05). However, no significant difference in other lipid parameters was detected in the baseline. After 6 months of LT4 therapy, the lipidemic profile in our patients with subclinical hypothyroidism did not change significantly. Thus, the limited improvement in the lipidemic profile of our patients may partly explain why the serum FGF-21 levels were increased after 6 months of LT4 treatment, but this increment was not sufficient to prove statistical significance. A longer-term follow-up of our patients after LT4 initiation could provide further evidence.

In this study, serum FGF-21 levels were associated with FT4 levels, but further analysis showed no actual interaction between these variables, indicating possible unknown confounding factors. It has been proposed that FGF-21 and THs may act both synergistically and independently [7,13,17,35]. In a previous animal study, the chronic infusion of FGF-21 significantly increased serum TSH, T3, and T4 levels [9]. In addition to these findings, Domouzoglou et al. [18] reported that T3 administration regulates FGF-21 transcription and increases circulating FGF-21 levels in animal models. On the other hand, a closer molecular look at the individual TH and FGF-21 pathways in those animal knock-out mice models revealed that distinct metabolic pathways are affected [18].

In a recent study in humans [15], a change in FGF-21 levels after LT4 treatment was well correlated with the increase in FT3 and FT4 values. Similarly, in our study, FGF-21 levels follow the same augmenting trend after LT4 treatment in children with subclinical hypothyroidism. However, it was not firmly established that there is a clear association between the rise in FGF-21 levels and the change in metabolic parameters in patients with hypothyroidism [12,15]. On the contrary, reports show a negative linear association between FT4 and plasma FGF-21 levels, even after multiple co-founders’ adjustments (sex, BMI, TG, and glucose) [14]. Thus, it seems that there is a crosstalk between the metabolic pathways that involve THs and FGF-21 that needs further elucidation.

In terms of glycemic parameters and FGF-21, several studies have yielded equivocal results. In our study, glucose levels, fasting insulin, and HOMA-IR in the patients with subclinical hypothyroidism were within the normal range and did not differ among controls and AIT euthyroid individuals, at baseline or after LT4 treatment. Our results are in accordance with the study of Lee et al. [14]. No difference in glucose levels was detected, and the observed change in plasma FGF-21 levels was independent of glucose metabolism. Similarly, in another study in adults, fasting glucose variation did not differ between groups at baseline, although a significant difference in HOMA-IR was reported. However, FGF-21 was not significantly correlated with HOMA-IR [15]. Hanks et al. [35] showed that FGF-21 was inversely associated with HOMA-IR in boys 7–12 years of age but not in girls. In another study in a pediatric population, no relation between FGF-21 levels and glucose and insulin levels changes during an oral glucose tolerance test was described [30]. More recently, Lei et al. [33] reported no significant baseline differences in fasting glucose and 2 h postprandial glucose levels among the hypothyroidism, euthyroidism, and controls groups. However, insulin and HOMA-IR values were lower in the young patients with hypothyroidism. A study with 70 children with obesity and 45 without obesity showed that FGF-21 levels were significantly correlated with HOMA-IR after adjusting for BMI, TG, HDL, and adiponectin levels [32].

Although the role of THs in RMR is well described [8,36,37], we were unable to detect a difference in RMR values between the studied groups, even after LT4 treatment. Furthermore, our study reported no significant correlation between serum RMR and FGF-21 levels after FFM and body weight adjustment. This finding could be explained by the fact that all participants presented with subclinical hypothyroidism and were promptly treated with LT4.

A well-described association exists between FGF-21, THs, and diet [11,38]. Our subjects did not follow any specific diet plan as part of the inclusion criteria. The overall KIDMED score was not significantly different between the studied groups, and it was not correlated with serum FGF-21 levels. However, the effect of diet on FGF-21 levels cannot be excluded, as healthy subjects in our study reported specific healthier eating habits (e.g., less junk food, fewer sweets, and more vegetables and nuts).

Finally, in the AIT treatment group, a notable increase in the FM and FFM levels after LT4 treatment was observed (*p* < 0.05), even after adjusting for age. That increment was independent of FGF-21 levels, the overall adiposity, and the BMI. These effects could be explained by restoring THs levels and their subsequent impact on orexigenic neuropeptides, leptin levels, and the hypothalamic-pituitary-thyroid (HPT) axis regulation [8]. In a previous study, no significant correlation between FGF-21 levels and DXA-derived fat percentage or BMI in healthy children has been detected [30], although more recently, an inverse association between FGF-21 levels and lean mass in girls was described, independently of FM [35]. On the other hand, in adults with hyperthyroidism, the serum FGF-21 levels have been negatively associated with %BF [16]. The different body composition assessment techniques that were used make the comparison of the above results challenging. At the same time, the biochemical pathways involved in this process should be further investigated.

The present study has some limitations that need to be recognized. The design of the study as a prospective cross-sectional protocol does not presuppose randomization; the sample size was relatively small. Serum FGF-21 levels were not measured in AIT euthyroid patients, as previous animal studies revealed that exogenous FGF-21 administration to hypothyroid animal models led to similar serum and liver lipid metabolites and gene expression changes in both hypothyroid and euthyroid mice [18]. Most of the participants did not have severe long-standing hypothyroidism before starting LT4 treatment, making it more difficult to detect the subtle, if any, metabolic changes that such pediatric patients develop.

## 5. Conclusions

To the best of our knowledge, this was the first attempt to study FGF-21 levels in relation to RMR and LT4 therapy in pediatric patients with Hashimoto’s thyroiditis. The present study found that serum FGF-21 levels are not significantly different between the healthy subjects and those with AIT and subclinical hypothyroidism. More specifically, serum FGF-21 levels are lower in children and adolescents with AIT and subclinical hypothyroidism compared to healthy controls, without reaching statistical significance. The decrease in serum FGF-21 levels in children and adolescents with AIT and subclinical hypothyroidism is not so evident as in the case of overt hypothyroidism due to Hashimoto’s thyroiditis. Serum FGF-21 levels tend to increase but not significantly, 6 months after LT4 treatment. Finally, LT4 therapy for 6 months has no apparent effect on RMR levels, lipid concentrations, or liver or glycemic parameters. Further studies with a larger number of young patients with severe hypothyroidism are needed to confirm the association between FGF-21 levels and thyroid function.

## Figures and Tables

**Figure 3 medicina-57-01374-f003:**
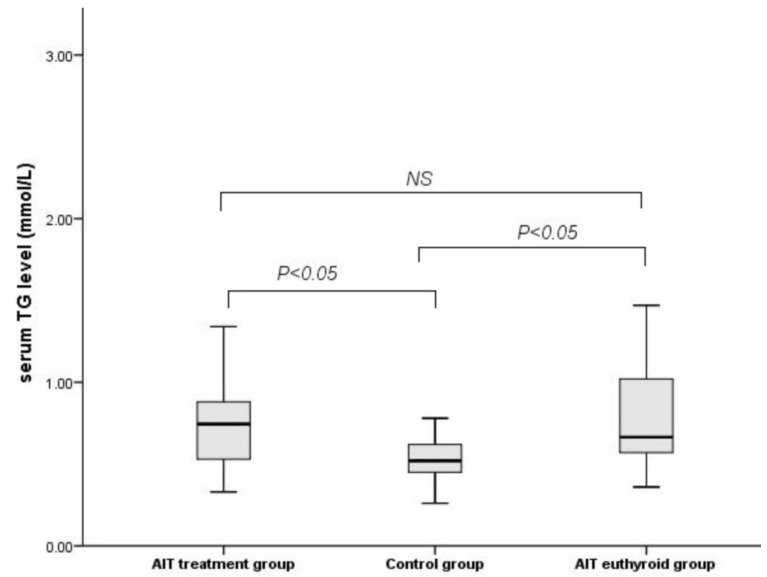
Serum TG levels of the control, AIT treatment, and AIT euthyroid groups. Values are expressed as median and range. TG: triglyceride; AIT: chronic autoimmune thyroiditis; NS: not significant.

**Table 1 medicina-57-01374-t001:** Baseline characteristics of all participants.

Parameter	Control Group (*n* = 30)	AIT Treatment Group (*n* = 30)	AIT Euthyroid Group (*n* = 30)	*p*-Value
Age (yrs)	10.89 ± 2.29	10.99 ± 1.85	11.01 ± 1.93	0.970
SDS BMI	0.88 (−0.01–1.22)	0.73 (0.12–1.64)	0.36 (−1.07–1.16)	0.046
WAIST C. (cm)	69.33 ± 9.99	67.73 ± 9.32	65.65 ± 8.59	0.313
HIP C. (cm)	80.63 ± 10.58	81.38 ± 9.58	77.41 ± 9.69	0.267
MUAC (cm)	23.00 (20.00–25.25)	22.00 (20.00–25.25)	21.00 (19.75–23.62)	0.195
%BF	22.6 (18.53–31.8)	24.95 (21.15–32.07)	21.65 (16.47–29.83)	0.482
FMI (kg/ht^2^)	4.42 (3.17–5.98)	4.49 (3.43–7.06)	4.12 (2.64–5.63)	0.313
FFMI (kg/ht^2^)	14.66 (13.43–15.31)	14.27 (13.50–14.63)	13.47 (12.69–14.58)	0.103
TSH (μIU/L)	2.40 (1.95–3.06)	4.82 (3.99–9.66)	2.63 (2.13–3.09)	0.002
FT3 (pmol/L)	6.28 (5.71–6.79)	5.96 (5.44–6.40)	6.27 (5.91–6.71)	0.295
FT4 (pmol/L)	15.19 ± 1.93	14.29 ± 2.45	15.19 ± 2.57	0.313
Glucose (mmol/L)	4.88 (4.65–5.11)	4.86 (4.66–5.12)	4.74 (4.49–5.05)	0.302
Insulin (pmol/L)	48.12 (37.36–81.32)	64.58 (46.46–86.87)	54.65 (33.96–81.87)	0.472
HOMA-IR	1.50 (1.11–2.54)	1.93 (1.39–2.72)	1.62 (1.02–2.52)	0.143
TC (mmol/L)	3.88 ± 0.75	4.22 ± 0.84	4.22 ± 0.69	0.144
TG (mmol/L)	0.52 (0.45–0.65)	0.74 (0.52–0.89)	0.66 (0.56–1.02)	0.042
HDL (mmol/L)	1.38 (1.23–1.63)	1.43 (1.18–1.65)	1.54 (1.24–1.99)	0.157
LDL (mmol/L)	2.07 (1.65–2.56)	2.30 (2.01–2.92)	2.33 (2.04–2.64)	0.501
AST (IU/L)	24.50 (18.75–28.25)	23.00 (19.00–25.25)	27.50 (23.75–29.00)	0.253
ALT (IU/L)	14.50 (13.00–18.00)	15.00 (13.00–19.50)	16.50 (13.75–19.00)	0.410
γ-GT (IU/L)	12.00 (10.00–13.25)	12.00 (10.75–15.00)	12.00 (11.00–13.25)	0.284
ALP (IU/L)	217.00 (149.5–275.50)	200.00 (157.75–291.50)	217.00 (183.25–275.00)	0.475
RMR/Weight (kJ/kg per d)	150.46 (122.09–190.58)	131.08 (108.62–165.10)	168.74 (133.26–193.51)	0.089
FGF-21 (pg/mL)	217.36 (193.60–235.21)	182.71 (169.32–234.55)	Na	0.717

na = not available data. Data are expressed as mean ± SD or median (upper and lower quartiles). *p* = significant difference between groups at *p* < 0.05. Statistics: ANOVA within the GLM function and Box-Cox transformation of the response variable to improve normality. AIT = chronic autoimmune thyroiditis, SDS = standard deviation score, BMI = body mass index, C. = circumference, MUAC = mid-upper arm circumference, %BF = body fat percentage, FMI = fat mass index (FM/ht^2^), FFMI= fat-free mass index (FFM/ht^2^), TSH= thyroid-stimulating hormone, FT3 = free triiodothyronine, FT4 = free thyroxin, HOMA-IR = homeostatic model assessment for insulin resistance, TC = total cholesterol, TG = triglyceride, HDL = high-density lipoprotein, LDL = low-density lipoprotein, AST = aspartate aminotransferase, ALT = alanine aminotransferase, γ-GT = gamma gloutamyltransferase, ALP = alkaline phosphatase, RMR = resting metabolic rate, FGF-21 = fibroblast growth factor-21.

**Table 2 medicina-57-01374-t002:** Correlation of various parameters with FGF-21 in all participants.

Parameter	Correlation Coefficient (*r*)	*p*-Value
%BF	0.015	0.887
FMI (kg/ht^2^)	−0.017	0.871
FFMI (kg/ht^2^)	−0.036	0.738
TSH (μIU/L)	−0.096	0.367
FT3 (pmol/L)	0.177	0.094
FT4 (pmol/L)	0.399	0.001 *
Glucose (mmol/L)	−0.035	0.741
Insulin (pmol/L)	0.078	0.463
HOMA-IR	0.083	0.436
TC (mmol/L)	−0.025	0.812
TG (mmol/L)	0.17	0.872
HDL (mmol/L)	0.089	0.405
LDL (mmol/L)	−0.23	0.833
AST (IU/L)	0.31	0.771
ALT (IU/L)	−0.035	0.742
γ-GT (IU/L)	−1.05	0.324
ALP (IU/L)	−0.060	0.576
RMR/FFM (kJ/kg per d)	0.088	0.411
RMR/Weight (kJ/kg per d)	0.064	0.551
KIDMED score	−0.200	0.126

* Significant at the 0.01 level. *p* = significant correlation at *p* < 0.05. Statistics: Spearman rank correlation. BF = body fat, FM = fat mass, FMI = fat mass index (FM/ht^2^), FFM = fat-free mass, FFMI = fat-free mass index (FFM/ht^2^), TSH = thyroid-stimulating hormone, FT3 = free triiodothyronine, FT4 = free thyroxin, HOMA-IR = homeostatic model assessment for insulin resistance, TC = total cholesterol, TG = triglyceride, HDL = high-density lipoprotein, LDL = low-density lipoprotein, AST = aspartate aminotransferase, ALT = alanine aminotransferase, γ-GT = gamma gloutamyltransferase, ALP = alkaline phosphatase, RMR = resting metabolic rate, KIDMED = Mediterranean diet index for kids, FGF-21 = fibroblast growth factor-21.

**Table 3 medicina-57-01374-t003:** Characteristics of patients with Hashimoto’s thyroiditis and subclinical hypothyroidism before and after 6 months of treatment.

Parameters	Before Treatment (*n* = 30)	After Treatment (*n* = 30)	*p*-Value
SDS BMI	0.73 (0.12–1.64)	0.88 (0.10–1.50)	0.915
WAIST C. (cm)	67.73 ± 9.32	69.2 ± 8.61	0.529
HIP C. (cm)	81.38 ± 9.58	81.50 ± 14.71	0.971
MUAC (cm)	22.00 (20.00–25.25)	23.00 (22.00–26.00)	0.215
%BF	24.95 (21.15–32.07)	24.06 (19.27–31.16)	0.955
FMI (kg/ht^2^)	4.49 (3.43–7.06)	8.98 (4.87–13.44)	0.001
FFMI (kg/ht^2^)	14.27 (13.50–14.63)	24.90 (18.52–31.31)	0.000
TSH (μIU/L)	4.82 (3.99–9.66)	4.03 (2.43–6.20)	0.054
FT3 (pmol/L)	5.96 (5.44–6.40)	5.90 (5.22–6.21)	0.055
FT4 (pmol/L)	14.29 ± 2.45	15.70 ± 2.45	0.037
Anti-TPOAb (IU/mL)	979.50 (135.27–2383.72)	800.00 (73.47–1687.20)	0.400
Anti-TgAb (IU/mL)	283.90 (69.40–500.00)	236.20 (87.95–500.00)	0.401
Glucose (mmol/L)	4.86 (4.66–5.12)	4.97 (4.72–5.19)	0.374
Insulin (pmol/L)	64.58 (46.46–86.87)	65.07 (47.85–87.43)	0.346
HOMA-IR	1.93 (1.39–2.72)	1.90 (1.57–2.65)	0.405
TC (mmol/L)	4.22 ± 0.84	4.19 ± 0.86	0.913
TG (mmol/L)	0.74 (0.52–0.89)	0.73 (0.56–0.93)	0.517
HDL (mmol/L)	1.43 (1.18–1.65)	1.46 (1.26–1.61)	0.865
LDL (mmol/L)	2.30 (2.01–2.92)	2.30 (1.93–2.82)	0.781
AST (IU/L)	23.00 (19.00–25.25)	22.00 (18.00–25.00)	0.096
ALT (IU/L)	15.00 (13.00–19.50)	15.50 (13.75–19.25)	0.315
γ-GT (IU/L)	12.00 (10.75–15.00)	11.00 (10.00–14.00)	0.652
ALP (IU/L)	200.00 (157.75–291.50)	224.50 (162.50–270.25)	0.414
RMR/Weight (kJ/kg per d)	131.08 (108.62–165.10)	142.51 (116.61–168.53)	0.517
FGF-21 (pg/mL)	182.71 (169.32–234.55)	198.43 (183.86–248.42)	0.734

Data are expressed as mean ± SD or median (upper and lower quartiles). *p* = significant difference between groups at *p* < 0.05. Statistics: ANOVA within the GLM function and Box-Cox transformation of the response variable and subject ID as a co-variate. SDS = standard deviation score, BMI = body mass index, C. = circumference, MUAC = mid-upper arm circumference, BF = body fat, FMI = fat mass index (FM/ht^2^), FFMI= fat-free mass index (FFM/ht^2^), TSH = thyroid-stimulating hormone, FT3 = free triiodothyronine, FT4 = free thyroxin, HOMA-IR = homeostatic assessment model for insulin resistance, TC = total cholesterol, TG = triglyceride, HDL = high-density lipoprotein, LDL = low-density lipoprotein, AST = aspartate aminotransferase, ALT = alanine aminotransferase, γ-GT = gamma gloutamyltransferase, ALP = alkaline phosphatase, RMR= resting metabolic rate, FGF-21 = fibroblast growth factor-21.

## Data Availability

The data presented in this study are available on request from the corresponding author. The data are not publicly available due to privacy and ethical reasons.

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
