# Peer review of "Serum Fibroblast Growth Factor 21 Levels in Children and Adolescents with Hashimoto’s Thyroiditis before and after l-Thyroxin Medication: A Prospective Study"

_medicina, 2021, doi:10.3390/medicina57121374_

Round 1

Reviewer 1 Report

An interesting study that assesses a possible relationship between patients with autoimmune thyroiditis in euthyroidism and primary hypothyroidism (Hashimoto's thyroiditis) and Fibroblast growth factor 21 (FGF-21), which shares common sites of action with thyroid hormones.

Some points should be highlighted:

1) Patients in the Hashimoto's thyroiditis (HT) group in euthyroidism should be classified as euthyroid with chronic autoimmune thyroiditis because a priori, HT means to present primary autoimmune hypothyroidism and this group had detectable anti-TPO and/or anti-thyroglobulin antibodies and suggestive ultrasound only.

Patients classified as HT hypothyroidism, according to table 1 and description of the series, had subclinical hypothyroidism or even were biochemically in euthyroidism, with clinical symptoms suggestive of hypothyroidism. It is known that the symptoms of hypothyroidism, especially in the subclinical, are nonspecific and, therefore, should not be considered for the classification of patients.

In my opinion, there is a bias in the classification of patient groups, and only patients with high TSH and free T4 below the reference values ​​should be classified as hypothyroid. Another group could be created: subclinical hypothyroidism.

2) FGF-21 was not evaluated in the control group, which can also be considered a bias. The authors found no difference between FGF-21 concentrations in the “euthyroidism” and “hypothyroidism” groups, perhaps because there is not really a group of patients in hypothyroidism or perhaps FGF-21 is linked to thyroid autoimmunity, a difference that could be found when comparing HT groups with healthy euthyroid groups without thyroid autoimmune disease.

3) In Results, lines 167 and 168, the authors described "The difference of FGF-21 serum levels between the hypothyroid and the healthy groups did not reach statistical significance (all p > 0.05)." But, in Table 1, the FGF-21 is described as “na= not available data” for the Euthyroid Group. Could the authors clarify this point?

4) When describing a result with no significant difference, it is recommended to refer that there was no difference. Several times the authors comment in results that one numerical variable was greater than another, but it was not significantly different, including in the conclusion of the abstract, for example “: In youngsters with hypothyroidism due to Hashimoto's thyroiditis, the serum FGF-21 levels are mildly lower than in healthy individuals and increase 30 after treatment with L-thyroxin without a statistical significance”. In my opinion, it should be described that there were no differences, as the way in which the description is presented may confuse the reader.

Similarly to what happened in lines 213 to 215: “Although serum FGF-21 levels correlated with the FT4 levels at baseline (r = 0.385, p < 0.05), the correlation weakened after 6 months of therapy (r = 0.246, p > 0.05 ).” The correlation was considered weak at baseline and it should be described that there was no correlation after LT4 replacement.

5) Throughout the Discussion, authors need to review how they refer to the data, strictly following the results found. For example, in lines 238 to 240: “In this prospective study, serum FGF-21 levels were mildly lower at baseline in children and adolescents with HT and hypothyroidism, and they were measured moderately increased after 6 months of LT4 therapy. FT4 concentrations correlated well with FGF-21 levels”. It is necessary to review the whole discussion, including the conclusions.

Reviewer 2 Report

The work is clinically impressive: very complete to try to measure peripheral action of thyroid hormones on basic metabolic rate, lipid metabolism, glycemic metabolism, liver function and FGF21.

I am not sure that your statistical approach is adequate. If I understand well, you have used a General Linear Model to take in account the variation due to the set of all demographic, physiological and biochemical variables. By this approach, you show an absence of "effect" of thyroid hormones on FGF21 level. This approach is not adequate: with a small number of subjects (30 in each group), you will always submerge an effect within his covariables. I suggest you to use a more classical approach for your data and you will see that the effect of thyroid status on FGF21 is present. FGF21 is thus a good marker of peripheral thyroid action according to your work. Is it better than triglyceride? A ROC curve comparing AUC sensitivity/of triglyceride versus FGF1 could bring interesting results. So, the data are interesting, but the way to analyse them should be reviewed. 

Other remarks/suggestions:

  1. Entire manuscript: HT : I had to read twice the introduction to realize that HT does not refer to hypo- or hyperthyroidism, but to Hashimoto disease. So, please, change this abbreviation to AIT, a more classical acronym for this syndrome;
  2. Table 1: if you accept the first point hereabove, it becomes more readable to define the groups as following:
    Control (no AIT, normal thyroid function) (n = 30)
    AIT with biochemical hypothyroidism before Treatment (n = 30)
    AIT with biochemical euthyroidism (n = 30)
  3. Statistic analysis:
    When comparing three groups as in table review 1, a more classical approach is to use oneway ANOVA test and post-hoc if oneway anova statistically significant unpaired Student t-test comparing two by two the three groups (A vs B; B vs C; A vs C). When using this approach for example for the variable FFMI, I obtain different results with a P < 0,0001 !

0neway ANOVA analysis for FFMI

No subjects

Mean

Standard deviation

Control group

30

14,66

1,23

AIT without biochemical hypothyroidism

30

14,27

0,77

AIT with biochemical hypothyroidism before treatment

30

13,47

0,78

P  <0,0001

Student t test for FGF21

Control group

30

217,36

23,76

AIT without biochemical hypothyroidism

30

182,71

13,39

P<0,0001

  1. For the clarity of the readers, it would be much more easy to use such a classical oneway ANOVA test for variables with normal distribution. Of course, as there are many parameters, multiparametric correction has to be added such as Bonferroni correction.
  2. Table 144 last line: at minimum, you should give an explanation in the “Discussion” section on the limitation of your work due to the absence od measurement of FGF21 in the control group. I guess there was pragmatic reason for this, but it is a “major” limitation of your work.
  3. Figure 2: it is a typical presentation of oneway ANOVA with post-hoc unpaired two*two. Student’t test. So, please, review your “statistical analysis” of Table 1 and of Methods section to be coherent.
  4. For categorical comparisons when values are not normally distributed, an OR Odd ratio measurement and X2 test would be much more readable than medians and interquartile range.

Round 2

Reviewer 2 Report

Comment(2) to Response to Reviewer 2 Comments

OK with your reponse. One remark remains:

Point 1 follow-up: suggestion to the author: one of the valuable information of your manuscript is that your results suggest that FGF21 could be a possible marker of peripheral hypothyroidism – even if statistical significance is not reached with the sample size used.       Actually, you have all the data to rapidly test a ROC presentation with SPSS: for example, binary variable: “AIT with normal thyroid function (n=30)” vs “AIT with abnormal thyroid function (n=30)”; continuous values of FGF21 and TG levels ordered in order of “1-specificity” and “sensitivity”. This kind of presentations should significantly complement your description by answering these questions:
a) Are TG and FGF1 ROC curve far from the diagonal ? AUC?    
b) Do AUC TG and AUC FGF1 differ?

In my opinion, this kind of info could significantly add information to your manuscript and to the Journal. Of course, you decide.   

  • Example of ROC curve for HbA1cand Fasting Plasma Glucose to detect newly
    diagnosed diabetes defined by OGTT – oral glucose tolerance test in patients with
    coronary artery disease. By analogy: ROC curve for TG and FGF1 in hypothyroidism defined by thyroid function test normal (AIT with normal thyroid function) or abnormal (AIT with abnormal thyroid function tests).
    SO: Wang JS, Lee IT, Lee WJ, et al. Performance of HbA1c and fasting plasma glucose in screening for diabetes in patients undergoing coronary angiography. Diabetes Care. 2013;36(5):1138-1140.
